# HYPERPARAMETER SEARCH ON THE TEST SET IN THE WILD

## ABSTRACT

Systems neuroscience has rapidly adopted machine-learning techniques, but has yet to develop a robust standardized methodology for assessing the performance of decoding models. Methodological issues can sometimes be subtle, arising as a consequence of experimental design. Here, in contrast, we investigate the consequences of post-hoc model selection: an issue which is neither subtle nor idiosyncratic. This occurs when a single test set is used to both select hyperparameters and evaluate performance, which favors models that overfit to ungeneralizable features of the test set. While the issues with this practice have been well documented within the ML literature, it has seen continued use in several domains, including systems neuroscience. To highlight this unfortunate practice, we performed a series of experiments using a selection of models from affected EEG decoding studies, finding that the overestimation of decoding accuracy in the affected studies was substantial: ranging from 0.74–1.24%. Moreover, we demonstrate that post-hoc model selection favors unstable model architectures, as the variability in their performance increases the likelihood that an instance of the model will coincidentally match the test set. Comparisons of model performance under post-hoc model selection may thus mislead researchers into developing increasingly complex and unstable models which fail to outperform simpler, more stable, ones.

## 1 INTRODUCTION

In supervised machine-learning tasks, such as classification or regression, a model is trained on a training set and then evaluated on a separate test set. However, when seeking a high-performing model, it is often desirable to perform *model selection*, by considering a range of values for hyperparameters, such as the learning rate, weight decay, or the number of training epochs. Model selection is generally performed by using (a) validation set(s) to determine the most effective hyperparameters for a model, and then evaluating the performance of a model trained with those hyperparameters on an independent test set. However, in *post-hoc model selection*, there is no validation set, and instead, all models are evaluated on the same test set, and the decoding accuracy is reported as the accuracy of the most performant model.

It may not be immediately obvious why this practice is problematic, as each model is evaluated on data not used during training. However, as the number of models that are evaluated increases, the more likely it is that the best-performing model has overfit to ungeneralizable features of the test set. By way of analogy, suppose one wished to investigate the existence of precognitive abilities in humans by testing how often subjects can correctly predict the outcome of a series of fair coin tosses. As Fig. 1 illustrates, the more subjects that are tested, the more likely it is that one will randomly guess a high percentage of the outcomes correctly. If we selected the subject with the most correct guesses and consider their performance in isolation, the evidence for their precognitive abilities would appear compelling. However, the anomalous accuracy is simply a consequence of sample size, and if the subject repeated the experiment, we would expect them to perform at chance. The same principle applies to post-hoc model selection. As more models are evaluated, the likelihood that the best-performing model has overfit to ungeneralizable features of the test set increases. In

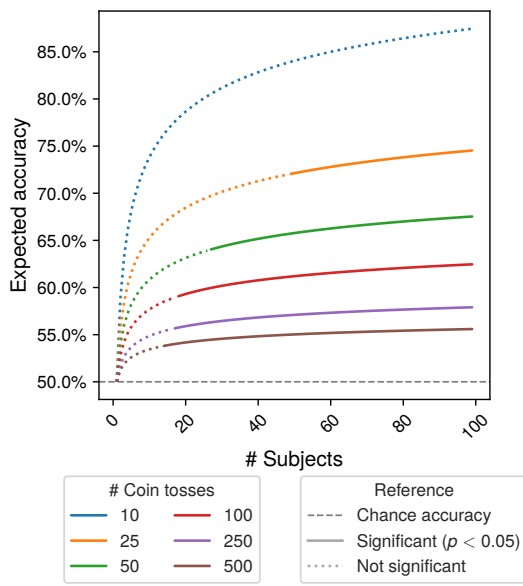

**Figure 1: Precognition or post-hoc selection?** Suppose a group of subjects are asked to predict the outcome of a series of fair coin tosses. The larger the group, the more likely it is that at least one subject will correctly guess a high proportion of the outcomes. For example, if there are 25 coin tosses to predict, we would expect any given subject to guess roughly 13 of the tosses correctly. However, as the plot illustrates, if there were 50 subjects, then we would expect at least one subject to guess 17/25 tosses correctly ($approx 70\%$). Moreover, a right-tailed binomial test would tell us that 17 or more correct guesses is statistically significant at the 95% confidence level. Nevertheless, it would be folly to conclude that the best-performing subject has pre-cognitive abilities. Similarly, in post-hoc model selection, the more models that are evaluated on a test set, the less likely it is that the estimated accuracy of the best-performing model is representative of its true decoding ability.

fact, for given any classification task, and an arbitrarily high desired accuracy, a sufficiently large number of random models can always be found such that the expected accuracy of the best-performing model exceeds the desired accuracy. See the supplementary material for the corresponding proof of this property.

While the potential for post-hoc model selection to bias performance estimates is a well-known issue, its continued use can likely be attributed to the perception that the overestimation of decoding accuracy is negligible given a test set of sufficient size. Moreover, if the emphasis of a study is on demonstrating a relative improvement in decoding accuracy, then it might be thought that the relative performance of models is still meaningful, even if the absolute performance is overestimated. However, as our analysis will show, neither of these assumptions hold. In particular, we demonstrate that post-hoc model selection can result in an overestimation of decoding accuracy which is not only statistically significant, but also of considerable practical relevance. Furthermore, we show that it does not affect all models equally, but imparts a greater bias to unstable model architectures, as the variability in their performance increases the likely extent to which the best-performing instance of a model architecture has overfit to the test set. And since a primary goal of a decoding study is often to demonstrate the efficacy of a new feature-engineering technique or model architecture, the publication of results which favor unstable models may encourage the development of increasingly large, complex, and unstable models which do not necessarily outperform simpler models.

The prevalence of post-hoc model selection within systems neuroscience is difficult to establish. However, while investigating a separate methodological issue known as the *repeated-stimulus confound*, Kilgallen et al. (2025) observed that, of the 18 studies identified as vulnerable to the confound, 11 were found to evaluate decoding performance via post-hoc model selection (Bagchi & Bathula, 2021; 2022; Luo et al., 2023; Kalafatovich et al., 2020; 2023; Kalafatovich & Lee, 2021; Fares et al., 2020; Jiang et al., 2021; Xue et al., 2024; Ahmadieh et al., 2023; 2024). Moreover, two additional publications feature models which require some form of hyperparameter tuning, but describe no model-selection procedures (Zheng et al., 2020; Deng et al., 2023). Therefore, it is possible that post-hoc model selection was performed implicitly in these studies by repeating the experiments using different hyperparameter values until a satisfactory decoding accuracy was achieved.

**Table 1:** Models from affected studies included in our experiments, with reported accuracies.

| Publication | Model | Concept-decoding[*] | Stimulus-decoding[*] |
|---|---|---|---|
| Kalafatovich et al. (2020) | ADCNN | 50.37%[†] | 26.75% |
| Bagchi & Bathula (2021) | AW1DCNN | 51.29%[†] | 28.68% |
| Bagchi & Bathula (2022) | CT-Slim | 51.96%[†] | 26.08% |
| | CT-Fit | 52.17%[†] | 27.14% |
| | CT-Wide | 52.33%[†] | 29.44% |
| Deng et al. (2023) | RLSTM | 52.69%[†‡] | 29.92%[‡] |
| Kalafatovich et al. (2023) | TSCNN | 54.28%[†] | — |
| Luo et al. (2023) | STST | 54.82%[†] | 29.98% |

'*' The accuracy reported in the corresponding publication.

'†' Obtained under an evaluation procedure susceptible to the repeated-stimulus confound.

'‡' No model selection method was reported.

## 2 MATERIALS AND METHODS

### 2.1 DATA AND DECODING MODELS

To perform our experiments, we used the Stanford University dataset (SUD; Kaneshiro et al. 2015), as well as a selection of decoding models from studies previously identified by Kilgallen et al. (2025) as performing post-hoc model selection. However, the reproducibility of models from the affected publications was limited in several instances. We also elected to include the model from Deng et al. (2023), as it is possible that post-hoc model selection was performed implicitly in that study. The models selected for use in the experiments, and their reported accuracies, are detailed in Table 1.

To demonstrate that post-hoc model selection favors unstable models, we also performed an additional set of experiments using logistic regression. While introducing stochasticity during training can be a legitimate technique for improving model generalization, we decouple instability from training by injecting controlled noise only at evaluation time. Specifically, we train a standard logistic-regression model $z(x) = Wx + b$, which we refer to as the stable baseline. Then, at evaluation time, we sample randomized decisions using the Gumbel-Max trick (Gumbel, 1954; Huijben et al., 2022)

$$\hat{y} = \arg\max_k \left\{ z(x) + \beta \left( -\log(-\log U_k) \right) \right\} \qquad U_k \overset{\text{iid}}{\sim} \text{Uniform}(0, 1) \qquad (1)$$

where $\beta > 0$ controls the instability of the model's predictions. By construction, as $\beta \to 0$, the procedure converges to the stable baseline, while an increase in $\beta$ raises the probability that a different class is predicted. To reliably control instability, we use a fixed weight decay, as weight decay shrinks margins and thereby affects the probability that a different class will be predicted. Additionally, to ensure that our findings are relevant to real-world contexts, in all experiments, a separate model instance is trained for each value of $\beta$.

### 2.2 MODEL TRAINING, SELECTION, AND EVALUATION PROCEDURES

To investigate the effects of post-hoc model selection in different contexts, we performed experiments for three distinct decoding tasks. Firstly, a 6-class concept-decoding task, where the objective is to predict the concept labels of responses to unseen stimuli. Secondly, the 6-class confounded concept-decoding task

**Table 2:** Accuracy by decoding task, model and selection method.

| Model | Concept decoding | | RSC Concept decoding | | Stimulus decoding | |
|---|---|---|---|---|---|---|
| | Pre-hoc | Post-hoc | Pre-hoc | Post-hoc | Pre-hoc | Post-hoc |
| **Affected models** | | | | | | |
| ADCNN | 43.18 | 44.40 | 51.37 | 52.37 | 32.76 | 33.49 |
| AW1DCNN | 41.57 | 43.02 | 49.21 | 50.28 | 13.47 | 14.06 |
| CT-Slim | 40.99 | 42.34 | 48.75 | 49.99 | 22.60 | 23.30 |
| CT-Fit | 42.32 | 43.39 | 50.27 | 51.03 | 20.35 | 21.13 |
| CT-Wide | 42.52 | 43.56 | 50.53 | 51.27 | 20.50 | 21.33 |
| RLSTM | 40.76 | 41.95 | 47.91 | 48.77 | 18.81 | 19.37 |
| TSCNN | 41.38 | 42.76 | 48.44 | 49.61 | 24.75 | 25.52 |
| STST | 38.81 | 40.09 | 44.91 | 45.84 | 21.85 | 22.53 |
| **Additional models** | | | | | | |
| LR | 37.74 | 38.01 | 43.96 | 44.36 | 12.98 | 13.12 |
| Unstable-LR | 37.56 | 38.38 | 43.87 | 44.64 | 12.93 | 13.36 |

performed in the affected publications, where the aim is to decode concept labels from unseen responses to previously seen stimuli. Lastly, a 72-class stimulus-decoding task, where the goal is to decode stimulus labels from unseen responses.

For all decoding tasks, we performed 12-fold nested cross validation with 11 inner folds. For each outer fold, one fold is used as the test set while the remaining 11 folds form the training set. The training set of each outer fold is then divided into 11 inner folds. Each inner fold is used as a validation set once while the remaining 10 inner folds form the training set. For each decoding task, we evaluated all hyperparameter combinations on both the inner and outer folds. Under post-hoc model selection, the hyperparameters which maximized accuracy on each outer fold were selected. And conversely, under pre-hoc model selection, the hyperparameters which maximized accuracy on the validation sets of each outer fold were selected. Details of the hyperparameter ranges used in our experiments are included in the supplementary material.

In our concept-decoding experiments, we extended the paired cross-validation procedure used in Kilgallen et al. (2025) to facilitate both pre-hoc and post-hoc model selection. This method had the added advantage of allowing us to quantify the bias imparted by post-hoc model selection both in isolation and in conjunction with the repeated-stimulus confound. See the supplementary materials for a pseudocode implementation of the cross-validation algorithm.

## 3 RESULTS

Table 2 presents the results of our decoding experiments. For each decoding task and model, we report the mean accuracy under both pre-hoc and post-hoc model selection. The concept-decoding and stimulus-decoding tasks were performed to investigate the impact of post-hoc model selection under two common decoding paradigms. As the affected publications we investigate in this work are also affected by the repeated-stimulus confound (RSC), the RSC concept-decoding task was performed to establish a comparison with the results reported in the affected publications. However, while we estimate the RSC concept-decoding accuracy of ADCNN as approximately 2% higher than was reported, the remaining models achieved accuracies below the reported values. The discrepancy was moderate for AW1DCNN, CT-Slim, CT-Fit, and CT-Wide ($<2\%$), more severe for RLSTM and TSCNN (3–5%), but worst for STST ($\approx 9\%$).

**Table 3:** Selection bias by task and model.

| Model | Concept decoding | RSC concept decoding | Stimulus decoding |
|---|---|---|---|
| **Affected models** | | | |
| ADCNN | $1.22^{***}$ | $1.00^{***}$ | $0.73^{***}$ |
| AW1DCNN | $1.44^{***}$ | $1.06^{***}$ | $0.58^{***}$ |
| CT-Slim | $1.36^{***}$ | $1.24^{***}$ | $0.70^{***}$ |
| CT-Fit | $1.07^{***}$ | $0.76^{***}$ | $0.78^{***}$ |
| CT-Wide | $1.04^{***}$ | $0.74^{***}$ | $0.82^{***}$ |
| RLSTM | $1.19^{***}$ | $0.87^{***}$ | $0.56^{***}$ |
| TSCNN | $1.38^{***}$ | $1.17^{***}$ | $0.77^{***}$ |
| STST | $1.28^{***}$ | $0.93^{***}$ | $0.67^{***}$ |
| **Additional models** | | | |
| LR | $0.27^{***}$ | $0.39^{***}$ | $0.14^{*}$ |
| Unstable-LR | $0.81^{***}$ | $0.77^{***}$ | $0.43^{***}$ |

'\*', '\*\*', and '\*\*\*' indicate that the bias due to post-hoc model selection is greater than 0 at the $p < 0.05$, $p < 0.01$, and $p < 0.001$ significance levels, respectively.

## 4    DISCUSSION

While it can be readily observed that accuracy is greater under post-hoc model selection than pre-hoc model selection for all models and tasks, further analysis is required to assess the significance of the observed bias.

### 4.1    POST-HOC MODEL SELECTION OVERESTIMATES DECODING ACCURACY

One factor that may contribute to the continued use of post-hoc model selection is the perception that the magnitude of the bias is relatively modest in practice. However, we empirically refute this rationale.

To support our claim, we performed hypothesis tests to determine if the bias affecting each model is significantly greater than zero, by applying one-tailed t-tests with confidence level $\alpha = 0.05$. At the task level, to account for the separate hypothesis tests performed for each model, we applied the Holm-Bonferroni procedure to adjust the significance levels of the tests (Holm, 1979). The results of the hypothesis tests are presented in Table 3.

Our findings indicate that, for each task, the bias in decoding accuracy due to post-hoc model selection is significantly different from zero for all models. Consequently, we conclude that selecting the hyperparameter combination which results in the highest test-set accuracy results in an overestimation of decoding accuracy. In fact, as Fig. 2 illustrates, the estimated accuracy of the model with the selected hyperparameters is not just optimistic, it is generally overestimated by a greater margin than that of any other model. Moreover, the substantial magnitude of the bias we observed indicates that it is not just statistically significant, but of practical relevance. For instance, it can be observed from Table 1 that, for the confounded concept-decoding task, the margin of improvement in each successive publication ranges from 0.36–1.59%, while our corresponding estimates of bias range from 0.74–1.24%. This may explain why such a high-proportion of studies which use the SUD also perform post-hoc model selection. In the absence of the bias conferred by post-hoc model selection, the margin by which a model would need to outperform the current state-of-the-art is effectively double the historical trend. Consequently, once a study which uses post-hoc model selection is published, it is difficult for subsequent work to revert to a more robust practice.

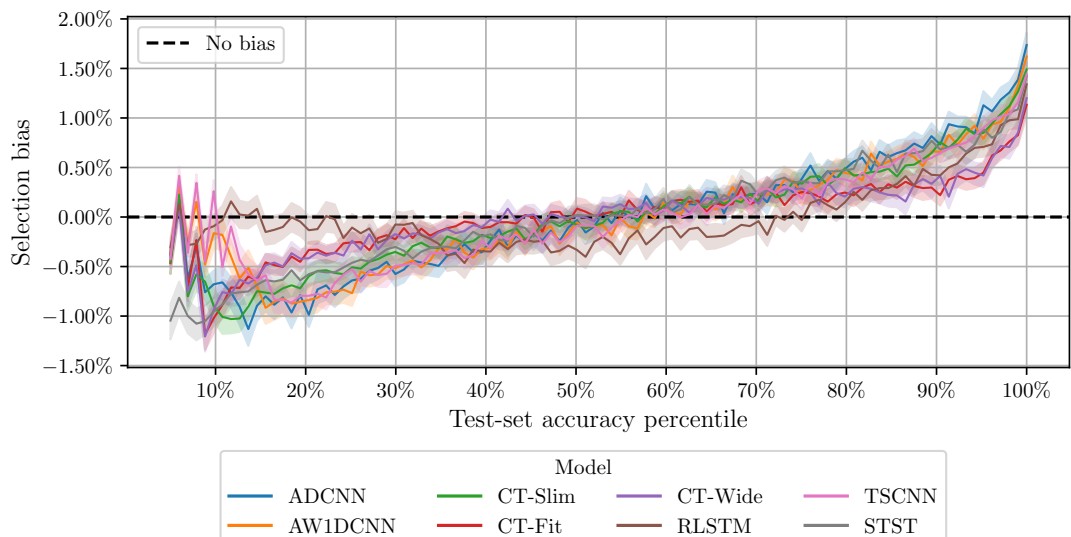

**Figure 2: Hyperparameters selection on the test set is cherry-picking, just look at the rest of the fruit.**
For each model, subject, and fold in our concept-decoding experiments, we estimated the selection bias for
the hyperparameter combination with the $i$th best accuracy on the test set by subtracting the test-set accuracy
of the hyperparameter set with the $i$th best validation-set accuracy. It can be clearly seen that, for all models
except RLSTM, the top 40–50% of all hyperparameter combinations overestimate accuracy. Moreover,
the magnitude of the overestimation is proportional to the relative performance on the test set. Therefore,
selecting model hyperparameters using the test set, is effectively cherry-picking the most optimistic, but
least reliable, estimate of model performance.

## 4.2 POST-HOC MODEL SELECTION OVERESTIMATES THE SIGNIFICANCE OF INCREASES IN DECODING ACCURACY

Another potential explanation for the continued use of post-hoc model selection is the assumption that,
despite any overestimation of accuracy, if it is standard practice, then relative improvements in performance
are still meaningful across models or studies.

Therefore, to dispel any notions that two wrongs make a right, we demonstrate that, under post-hoc model
selection, the apparent significance of relative improvements in accuracy is unreliable. To this end, for our
concept-decoding and stimulus-decoding tasks, we fit a linear mixed-effects model to estimate the accuracy
of each model. For both tasks, separate linear mixed-effects models were used to estimate accuracy under
pre-hoc and post-hoc model selection. Subsequently, post-hoc analysis was used to estimate the contrast
between different models using the Holm (1979) method. See the supplementary material for additional
information on the linear mixed-effects model used to construct the confidence intervals.

Fig. 3 depicts the 95% confidence intervals of the contrasts for each task and model-selection method.
With respect to the concept-decoding experiments, under post-hoc model selection, almost all contrasts are
deemed statistically significant, while the more robust procedure would indicate otherwise. However, this
finding was not duplicated in the stimulus-decoding experiments. We attribute this discrepancy to the nature
of the decoding tasks. In our concept-decoding experiments, each test set is composed of responses to
different unseen stimuli, while the test sets of the stimulus-decoding experiments all consist of responses to

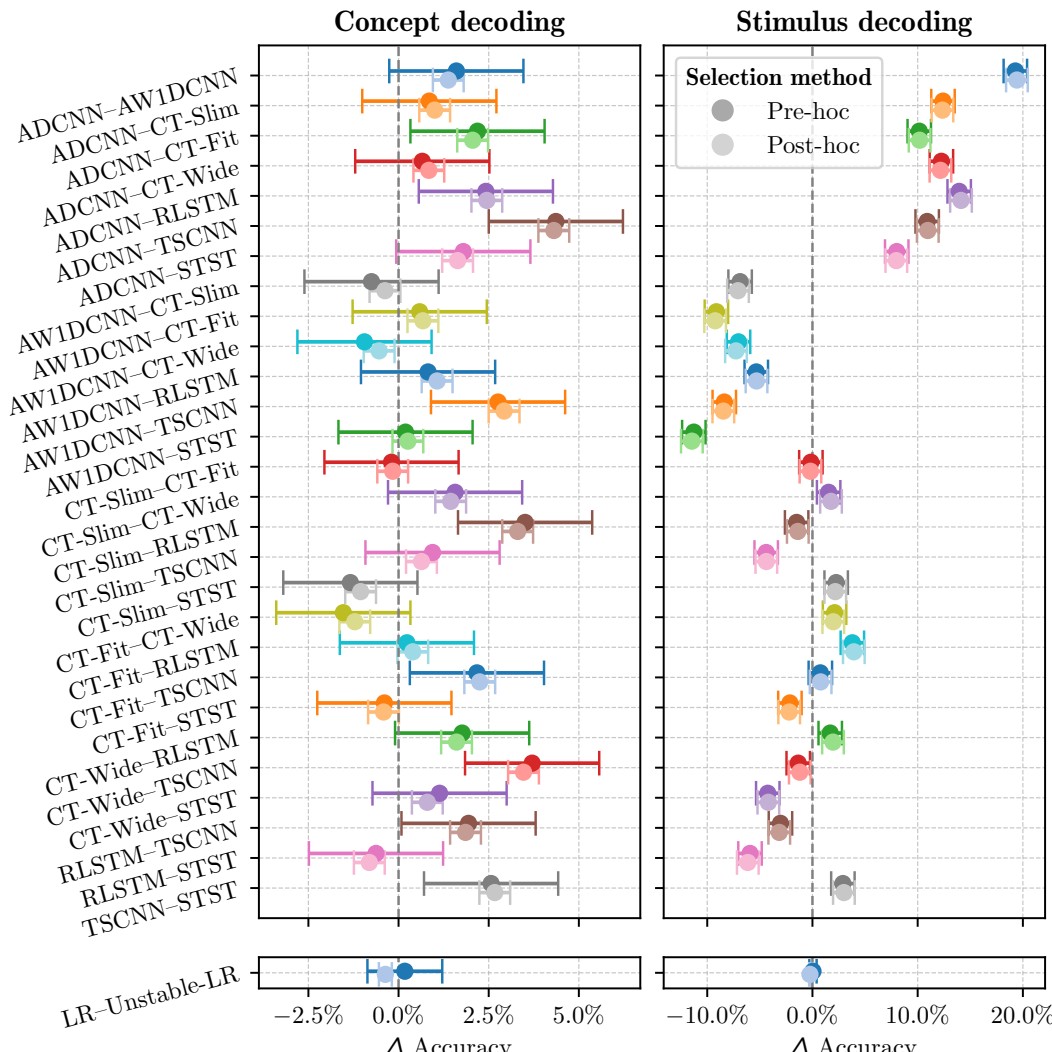

**Figure 3: The illusion of progress.** For each pair of models, the 95% confidence interval for the relative difference in decoding accuracy is depicted. An interval which does not include 0 indicates that the estimate is different from zero with statistical significance. In our concept-decoding experiments, when model hyperparameters are selected using a separate validation set, the difference in accuracy is statistically significant for only 10/28 pairs of models. However, when hyperparameters are selected using the test set, this increases to 26/28 pairs of models. Therefore, although each study claims that it establishes a new state-of-the-art by outperforming prior solutions, the validity of this claim is questionable. However, in the stimulus-decoding experiments, our hypothesis tests suggest that the only point of disagreement between the two model-selection procedures is that LR–Unstable-LR is only statistically significant under post-hoc model selection.

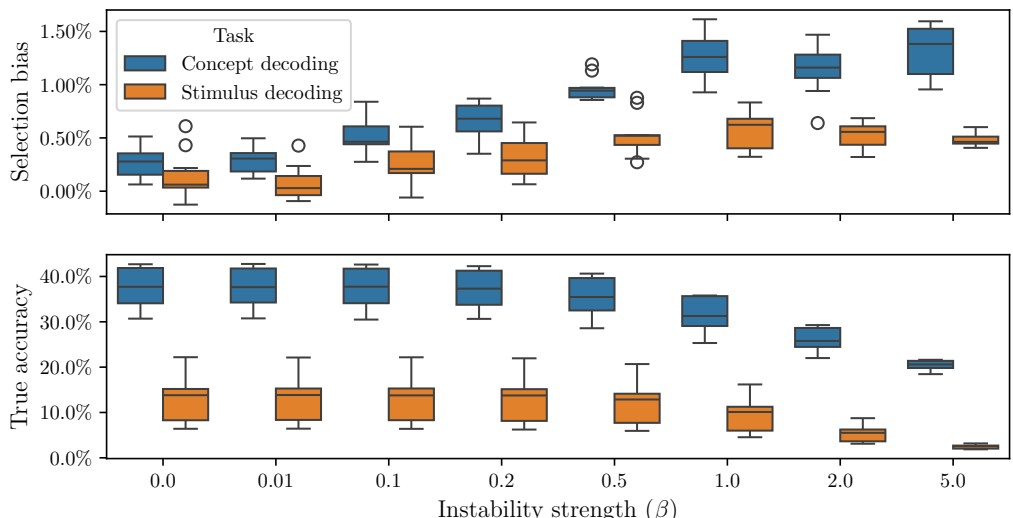

**Figure 4: Why bother with veracity when volatility is king?** The more unstable a model is, the less accurate it is, but the greater its accuracy is overestimated under post-hoc model selection. So, if a key aim of a decoding study is to achieve a new state-of-the-art, then the path of least resistance is to develop a less stable model of approximately equivalent decoding ability.

the same stimuli. Consequently, we observe that post-hoc model selection may be particularly brittle when a decoding task which is designed to capture the ability of a model to generalize to new stimuli.

### 4.3 POST-HOC MODEL SELECTION FAVORS UNSTABLE MODELS

The last potential motivation we offer for the continued use of post-hoc model selection is the belief that, regardless of its technical incorrectness it is a relatively harmless practice in the broader context of the literature. However, we present evidence which contradicts this hypothesis.

As mentioned previously, in addition to the models from publications known to perform post-hoc model selection, we also included two additional models, a conventional logistic-regression model, and a variant we designed to have controllably unstable predictions at evaluation time. It can be observed in Table 2 that, in all tasks, pre-hoc model selection favors conventional logistic regression, while the unstable variant is preferred under post-hoc model selection. Moreover, as Fig. 3 illustrates, post-hoc model suggestion suggests that our unstable logistic-regression model constitutes a statistically significant improvement over conventional logistic regression in both decoding tasks. However, as Fig. 4 illustrates, as instability increases, decoding accuracy decreases while selection bias increases.

The implications of this finding are far more insidious than it might appear. Model development is often an iterative process, where techniques like ablation studies are used to determine the aspects of a network which improve accuracy. However, since the overestimation of accuracy is proportionate to instability, this process may implicitly encourage researchers to develop increasingly complex and unstable models, which fail to outperform simpler solutions. Moreover, it is essentially a prerequisite of the publication process that a study documenting a novel decoding model should demonstrate that it outperforms prior solutions on some benchmark dataset. As a result, over time, the literature is likely to favor increasingly unstable models. While this may sound alarmist, it should be noted that the unstable logistic-regression model outperformed

its stable counterpart by a wider margin than RLSTM was reported to outperform CT-Wide (0.37% vs 0.36%).

Moreover, as novel decoding models may draw inspiration from prior solutions, the process as a whole may encourage the proliferation of techniques which result in more volatile, but less reliable, models. In addition to creating a misleading impression of progress, this issue may further contribute to the reproducibility crisis within systems neuroscience, as the more volatile a model is, the more difficult it is to reproduce a specific result. This may explain, in part, why the more recent a model is, the more our estimate of its performance differs from the concept-decoding accuracy reported in the corresponding publication.

## 5 CONCLUSION

In this work, we investigated the consequences of post-hoc model selection, a well-known but underestimated issue. We illustrated the theoretical nature of the problem using a thought experiment which highlighted that the same approach could also be used to support the existence of precognitive abilities in humans.

To demonstrate the severity of the issue in a real-world setting, we performed a series of decoding experiments using a selection of models from affected publications. Our analysis of the results revealed that the magnitude of the bias was both statistically significant and substantial for every model in all decoding tasks. Subsequent analysis also revealed two more subtle issues with the procedure. Firstly, that it substantially overestimates the significance of improvements in performance, and consequently creates a false impression of rapid progress and the relative efficacy of different decoding techniques. And secondly, it preferentially overestimates the accuracy of more unstable models, which may be implicitly encouraging researchers to develop increasingly complex and unstable models, which fail to outperform simpler solutions.

Given the consistent and substantial bias we observed, as well as the severe and varied nature of the other issues caused by post-hoc model selection, we suggest that its use should be immediately discontinued in favor of more robust model-selection methods. Moreover, the publication of studies which employ a practice known to be unsound indicates the need for both a more rigorous review process, and formal guidelines regulating how the results of machine-learning experiments are reported.

AUTHOR CONTRIBUTIONS

Removed for blind review.

ACKNOWLEDGMENTS

Removed for blind review.

ETHICS STATEMENT

This work debunks EEG analysis work based on faulty methods. Exposing these incorrect methods and consequent false results will allow resources wasted on continued use of these incorrect methods to be reallocated. The debunked work also actively causes harm, including grant proposals rejected due to preliminary results being uncompetitive with falsely-inflated performances based on faulty methods; manuscripts rejected for the same reasons; time wasted attempting to replicate the debunked results; and students learning invalid methods. Because the debunked work relates to brain-computer interfaces whose primary application is helping people with disabilities (e.g., paralysis) interact with the world, the harm is not merely scientific but also medical, with disproportionate affects on disabled people.

REPRODUCIBILITY STATEMENT

All data and code used to generate these results will be made publicly available following acceptance.

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

## A APPENDIX

### A.1 HOW POST-HOC MODEL SELECTION CAN BE USED TO ACHIEVE ARBITRARY ACCURACY

**Theorem 1** *Given any classification task, and an arbitrarily high desired accuracy, a sufficiently large number of random models can always be found such that the expected accuracy of the best-performing model exceeds the desired accuracy.*

**Proof 1** *We begin by observing that, for any classification task, the accuracy of a random model on a test set of $n$ samples follows some discrete probability distribution $D$ over a finite uniform grid $G = \left\{ \frac{i}{n} \right\}_{i=0}^{n}$ on $[0,1]$. Given a set of $m$ independent random models, the expected accuracy of the best-performing model, $A_m$, is given by*

$$A_m = \sum_{k=1}^{n} \frac{k}{n} \cdot \left[ F\left(\frac{k}{n}\right)^m - F\left(\frac{k-1}{n}\right)^m \right] \qquad (2)$$

*where $F$ is the cumulative distribution function (CDF) of $D$. And, since $F$ is a CDF over $[0,1]$, it holds that $F(x) \in [0,1] \ \forall x \in [0,1]$ and $F(x) = 1 \iff x = 1$. Therefore, it follows that, as $m$ goes to infinity, $F(1)^m$ dominates the sum, and thus:*

$$\lim_{m\to\infty} A_m = 1 \cdot \lim_{m\to\infty} F(1)^m = 1 \qquad (3)$$

*We can conclude from this that, for any $\epsilon > 0$, there exists $m \in \mathbb{N}$ such that $A_m > 1 - \epsilon$. Or equivalently, we can state that, for any classification task, under post-hoc model selection, evidence can always be found to suggest that a random model outperforms the state-of-the-art.*

### A.2 USING CLEVER CROSS VALIDATION TO QUANTIFY MODEL SELECTION BIAS

---

**Algorithm 1** Construction of paired cross-validation folds for the concept-level experiments.

---

**Input:** stimulus-grouped folds $\{\mathbf{idx}_i^{\alpha}\}_{i=1}^{S}$ and stimulus-stratified folds $\{\mathbf{idx}_i^{\beta}\}_{i=1}^{S}$, of trials indexed by $\mathbf{idx}$, where $S$ is the number of stimuli per category.
**Output:** Paired folds $\mathbf{idx}_{i,j,\bullet}^{\gamma}$ for $i \in \{1,\ldots,S\}$, $j \in \{1,\ldots,S\} \setminus \{i\}$, and $\bullet \in \{\lambda, \alpha, \alpha', \beta, \beta'\}$.

**for** $i \in \{1,\ldots,S\}$ **do**
    **for** $j \in \{1,\ldots,S\} \setminus \{i\}$ **do**
        $\alpha_{i,j} \leftarrow \mathbf{idx}_i^{\alpha} \cup \mathbf{idx}_j^{\alpha}$
        $\beta_{i,j} \leftarrow \mathbf{idx}_i^{\beta} \cup \mathbf{idx}_j^{\beta}$
        $\mathbf{idx}_{i,j,\alpha}^{\gamma} \leftarrow \mathbf{idx}_i^{\alpha} \setminus \beta_{i,j}$
        $\mathbf{idx}_{i,j,\alpha'}^{\gamma} \leftarrow \mathbf{idx}_j^{\alpha} \setminus \beta_{i,j}$
        $\mathbf{idx}_{i,j,\beta}^{\gamma} \leftarrow \mathbf{idx}_i^{\beta} \setminus \alpha_{i,j}$
        $\mathbf{idx}_{i,j,\beta'}^{\gamma} \leftarrow \mathbf{idx}_j^{\beta} \setminus \alpha_{i,j}$
        $\mathbf{idx}_{i,j,\lambda}^{\gamma} \leftarrow \mathbf{idx} \setminus \alpha_{i,j} \cup \beta_{i,j}$
    **end for**
**end for**
Given a fold $\mathbf{idx}_{i,j}^{\gamma}$, the training set, stimulus-unconfounded test and validation sets, and stimulus-confounded test and validation sets are indexed by $\lambda, \alpha, \alpha', \beta$ and $\beta'$, respectively.

---

**Table 4:** Hyperparameter ranges used in the experiments.

| Model | Hyperparameter | Range |
|---|---|---|
| ADCNN/AW1DCNN/RLSTM | learning rate | [0.0001] |
| | weight decay | [0.0001, 0.001, 0.01, 0.1] |
| | batch size | [64] |
| | # training epochs | $[1, 2, \ldots, 50]$ |
| CT-Fit/CT-Slim/CT-Wide | learning rate | [0.0001] |
| | weight decay | [0.0001, 0.001, 0.01, 0.1] |
| | $\gamma$ | [0.5] |
| | batch size | [64] |
| | # training epochs | $[1, 2, \ldots, 50]$ |
| | Projection origin | Electrode 51[†] |
| TSCNN | learning rate | [0.0001] |
| | weight decay | [0.0001, 0.001, 0.01, 0.1] |
| | batch size | [64] |
| | # training epochs | $[1, 2, \ldots, 50]$ |
| | $\tau_d$ | 0.2[‡] |
| | $\tau_f$ | 0.8 |
| STST | learning rate | [0.0001] |
| | weight decay | [0.0001, 0.001, 0.01, 0.1] |
| | batch size | [64] |
| | # training epochs | $[1, 2, \ldots, 50]$ |
| | wavelet | cmor($\Delta_f = 1.0, \ f_c = 1.0$) |
| LR | learning rate | [0.0001] |
| | weight decay | [0.0001] |
| | batch size | [64] |
| | # training epochs | $[1, 2, \ldots, 100]$ |
| Unstable-LR | $\beta$ | [0.01, 0.1, 0.2, 0.5, 1.0, 2.0, 5.0] |
| | learning rate | [0.0001] |
| | weight decay | [0.0001] |
| | batch size | [64] |
| | # training epochs | $[1, 2, \ldots, 100]$ |

'†' The marked value uses 1-based indexing.

'‡' Relative to the distance from nasion to inion.

### A.3    HYPERPARAMETER SELECTION

See Table 4.

### A.4    A POST-HOC ANALYSIS OF POST-HOC MODEL SELECTION

The linear mixed-effects model used to estimate the contrasts depicted in Fig. 3 is described by

$$Z_{i,j,k} = \beta_0 + \sum_{m=1}^{M} \beta_m \cdot M_i + s_j + \epsilon_{i,j,k} \qquad s_j \sim \mathcal{N}(0, \sigma_s^2) \qquad \epsilon_{i,j} \sim \mathcal{N}(0, \sigma^2) \qquad (4)$$

where $Z_{i,j,k}$ denotes the decoding accuracy of the $i$th model, on the $k$th fold of the $j$th subject. The intercept is given by $\beta_0$, and $\beta_m$ is the fixed effect for the $m$th decoding model. We use $M$ as a dummy variable for

decoding model, such that $M_x$ is equal to 1 if $i = x$, and 0 otherwise. The residual error and the random effect for the subject variable are denoted by $\epsilon$ and $s$ respectively. While the mixed-effects model technically only estimates the accuracy of each decoding model, post-hoc analysis was used to determine if any contrasts were significantly different from zero.

