# OpenReview forum: "Hyperparameter search on the test set in the wild"
_ICLR.cc/2026/Conference — Submitted to ICLR 2026_

### Official Review · Reviewer_uDoR · 2025-10-28

**Soundness:** 3
**Presentation:** 2
**Contribution:** 2
**Rating:** 4
**Confidence:** 3

**Summary:**

The paper tackle the problem of post-hoc model selection in system neuroscience where a validation set is not available. It provides an analysis of the bias of such post-hoc selection through a number of experiments.

**Strengths:**

- The paper addresses an important difficulty regarding hyperparameter optimisation and model selection when no validation set is present.
- The experiments and hyperparameters used in the experiments are explained in details.
- The paper presents insights about the flows of the post-hoc models and recommend replacing it with more robust model-selection methods.

**Weaknesses:**

- The concrete contribution of the paper is missing. While it can be derived from reading the whole paper, it is essential to explicitly present the contributions (ideally in the introduction). This helps any future readers to understand the contributions.
- The position of the research relative to related work is not clear. There is no “related work” section in the paper, making the position of the paper in relation to the existing research and whether it addresses a significant research gap unclear.
- While the paper provides important analysis and conclusions about the flaws of current method, it does not propose nor present any concrete solutions: The authors wrote the following: we suggest that its use should be immediately discontinued in favor of more robust model-selection methods”, which is a broad statement and does not present a concrete solution.
- The code and data are not available for the review process. The authors mention in the paper that “All data and code used to generate these results will be made publicly available following acceptance”. However, it would have been better to include an anonymized link for the reviewers to assess the code.

**Questions:**

In the appendix of the paper, Table 4 presents the exact hyperparameters ranges used in the experiments. However, the rationale behind selecting these exact values is missing. Can you please elaborate more on the process and rationale of setting these exact values? Did you consider using automated methods (e.g., AutoML) or were the values set based on something else?

---

> ### Author Response · Authors · 2025-11-21
> **Response to uDoR**
>
> Most of the main concerns and issues brought up have are addressed in the GENERAL COMMENTS above.
>
> ## W1
>
> We will add this; see GENERAL COMMENTS above.
>
> ## W2
>
> Also see above.
>
> ## W3
>
> See **Response to oCqR, §W2** above.
>
> ## W4
>
> Good point. We will try to do that in the future. If you'd like we can try to do that right now, and put the link here?
>
> ## Q1
>
> These were determined on a slightly ad-hoc basis, using information from the original cited studies where possible and appropriate, along with knowledge of the algorithm and data. An attempt was made to choose ranges which were reasonable, in terms of both generality and computational overhead.

---

### Official Review · Reviewer_LYdh · 2025-10-30

**Soundness:** 2
**Presentation:** 2
**Contribution:** 2
**Rating:** 2
**Confidence:** 4

**Summary:**

This paper raises the issue that post-hoc model selection can lead ML model performance to overfit test set features. Using an EEG dataset, the authors statistically demonstrate how models are affected by the problem(post-hoc model selection). They also point out that the problem can cause model overestimation, which in turn may lead to the development of unnecessarily complex and unstable model architectures.

**Strengths:**

- The paper highlights that post-hoc model selection can lead to the development of unstable model structures, which may negatively affect the design of suitable model architectures in the future.
- Through statistical tests, the authors identify biases caused by post-hoc model selection in deep learning models developed for EEG decoding tasks.

**Weaknesses:**

- The concept of post-hoc model selection is already a well-known issue. However, the related work section lacks a thorough review of prior studies that have addressed cherry-picking problems(post-hoc model selection).

- The paper identifies biases from post-hoc model selection using existing statistical tests, rather than developing a new methods. This work focuses more on demonstrating the severity of post-hoc model selection and suggesting the need for regulatory awareness. While the topic is critical, the paper may not fully align with the expectations of a main conference, which typically values methodological novelty. This work seems more suitable for a journal or review paper that focuses on ethical or methodological issues. If intended for a main conference, the paper would benefit from proposing a novel method to detect or mitigate post-hoc model selection effects, supported by mathematical justification and extensive experiments across multiple benchmarks.

- Although the paper title is “Hyperparameter Search on the Test Set in the Wild,” the experiments are conducted only on several models within an EEG decoding dataset. This issue is not unique to EEG decoding, yet the study does not examine diverse benchmarks or multiple model types. Therefore, the paper should either focus specifically on special domains like EEG decoding—where limited test sets make the problem particularly severe—and adjust the title accordingly, or broaden its scope to more general settings to strengthen its claims.

- The assumptions used in the appendix proofs do not appear sufficient to fully support the main argument. The results seem to rely on extremely small test sets and model performances that are already close to 1. In addition, a toy example demonstrating how these assumptions manifest in actual model outcomes would make the proof more convincing. At present, the assumptions seem simplistic, and if training does not converge properly, simply increasing the number of models may not guarantee convergence.

**Questions:**

- (W1) Are there no related works that have analyzed or addressed post-hoc model selection? Since cherry-picking is already a well-known issue, I assume that previous studies have examined this problem as well. In addition, this work appears somewhat similar to Kilgallen et al. (2025) [1], which introduced the repeated-stimulus confound. Could you elaborate on the differences and clarify the specific motivation behind their study?

- (W3) Why was only the EEG dataset used in this study? Was it chosen because such issues were observed in previous EEG decoding research? The problem of post-hoc model selection arises in performance evaluation across many ML applications, not just in EEG decoding. I would like clarification on whether the focus on EEG was due to practical limitations or deliberate scope.

- Was the t-test in Figure 3 performed on newly conducted post-hoc experiments, or based on existing model results? If the test set was used directly during experimentation, it would naturally lead to higher performance and thus significant results. What was the motivation behind conducting this analysis? Were there any specific observations or suspicions suggesting post-hoc model selection had occurred? In summary, it seems difficult to confirm post-hoc model selection solely from these results—what evidence led you to infer its presence?

- (W4) In the appendix, the
$F(x) \in [0, 1]\ \forall x \in [0, 1]\ \text{ and }\ F(x) = 1 \Leftrightarrow x = 1.$ — is this an assumption, or a general property of the CDF? It does not seem to hold as a general property of cumulative distribution functions, so is it a condition required to satisfy Eq. (3)?

[1] Kilgallen, Jack A., Barak A. Pearlmutter, and Jeffrey Mark Siskind. "The Repeated-Stimulus Confound in Electroencephalography." arXiv preprint arXiv:2508.00531 (2025).

---

> ### Author Response · Authors · 2025-11-21
> **Response to LYdh**
>
> Most of the main concerns and issues brought up have are addressed in the GENERAL COMMENTS above.
>
> ## W3
>
> Good point. We will change the title to add the specific domain.
>
> ## W4
>
> This was meant to be mainly illustrative. But we will try to weaken the assumptions and generalize this result. We can also add a toy example---good idea.
>
> ## Q3
>
> The analysis in Figure 3 is based on **newly conducted experiments** simulating both pre-hoc (using validation sets) and post-hoc (using the test set for selection) model selection procedures. These experiments used models from studies previously identified as performing post-hoc selection.
>
> **Post-hoc selection naturally led to greater accuracy** than pre-hoc selection for all models and tasks tested. This procedure was shown to **substantially overestimate the significance of increases in decoding accuracy**. For example, in concept-decoding experiments, the number of statistically significant differences between model pairs increased from 10/28 (pre-hoc) to 26/28 (post-hoc selection), creating a "false impression of rapid progress".
>
> The motivation was to investigate the negative consequences of post-hoc model selection in this domain, where this practice seems common.
>
> The evidence that led to the inference of the negative impact of this practice includes:
>
> 1. **Statistically Significant Bias:** The bias in decoding accuracy was found to be **significantly greater than zero for all models and tasks** tested. The magnitude of this bias (ranging from 0.74–1.24% in the confounded concept-decoding task) was substantial and comparable to historical margins of improvement reported in successive publications.
> 2. **Preference for Instability:** Post-hoc model selection preferentially overestimates the accuracy of more **unstable model architectures**. This finding suggests the practice may implicitly encourage the development of overly complex models that do not necessarily outperform simpler solutions.
>
> ## Q4
>
> The logic of the "F(x)=1 ⇔ x=1" in Appendix A.1 above Eq. (3) is indeed a bit off. The conclusion still holds, and we will fix the proof. (If the distribution has finite support over the entire interval then the statement does hold.)

---

### Official Review · Reviewer_oCqR · 2025-10-31

**Soundness:** 1
**Presentation:** 2
**Contribution:** 1
**Rating:** 2
**Confidence:** 4

**Summary:**

The current paper lays out the problem of post-hoc model selection in the context of EEG decoding, which is stated to be a well known problem in machine learning research but to be neglected in systems neuroscience. The authors propose three hypotheses for why this neglect occurs and attempt to test them using experiments with a subset of decoding models and tasks drawn from previous literature:
1. The magnitude of the bias caused by post-hoc model selection is perceived relatively modest in practice →  the paper shows that decoding accuracy substantially and consistently increases (i.e. positive selection bias) for all selected models and different decoding tasks for post-hoc compared to pre-hoc model selection, ranging from 0.74-1.24%.
2. There is the assumption that despite overestimation of accuracy due to post-hoc model selection, relative improvements in performance are still meaningful across models or studies → the paper shows that relative differences in decoding accuracy between models are more unreliable for post-hoc model selection than pre-hoc model selection within the context of concept decoding.
3. The belief that regardless of its technical incorrectness it is a relatively harmless practice in the broader context of the literature → using a direct manipulation of instability as implemented in a linear regression model, the paper shows that as instability increases, the selection bias increases, while true accuracy decreases.

Based on this, the authors claim that the process of post-hoc model selection favours unstable models, which might steer the field into the wrong direction for further model development and yields problems with reproducibility. Thus, the authors argue for the discontuition of post-hoc model selection and for more robust model-selection methods within EEG decoding research.

**Strengths:**

- The current paper adds to the existing literature by explicitly targeting the effects of post-hoc model selection in the context of EEG decoding on model comparison (on top of the effects of repeated stimulus confound).
- The paper systematically assesses the bias from post-hoc model selection for a diverse set of architectures and decoding tasks.
- It illustrates the implications of post-hoc model selection on model comparison through the modelling of instability in additional experiments.

**Weaknesses:**

Main weakness:
- The novelty and contribution of the paper appear limited. The issue of hyperparameter tuning on the test set has been well established in the machine learning literature [1, 2], and the current work primarily highlights this known problem within a non–machine learning context. Furthermore, the paper does not provide many new insights or concrete methodological recommendations for mitigating this issue. Given this, the contribution remains largely conceptual and does in my opinion not warrant acceptance.

The following points are (comparatively) more minor and could be addressed through revision. They aim to improve the clarity, rigor, and presentation of the paper:
1. It would be helpful to clarify the scope of the paper more explicitly. As it stands, the study appears to focus on EEG decoding methods aimed at optimizing performance for applied contexts such as brain–computer interfaces. This focus differs from research that uses decoding accuracy as a measure of representational similarity or discriminative distance in more fundamental neuroscientific investigations (e.g. [3], which also assesses different decoding architectures, preprocessing steps and data partitioning schemes within this context). Clearly delineating this scope would help readers understand the intended application domain. This distinction is currently also not reflected in the paper’s title.
2. The manuscript would benefit from a clearer discussion of the limitations of the current studies and more concrete recommendations for future research. For instance, while the authors demonstrate how post-hoc model selection can introduce selection bias, it remains somewhat unclear what specific methodological approach they recommend to mitigate this issue. Based on the methods described, nested k fold cross-validation appears to be implied, though this approach is relatively well-established and may not constitute a novel contribution.
3. The paper would benefit from greater specificity and a clearer logical structure in its writing. This issue manifests in several ways:
    - Introduction: The introduction currently concludes with a paragraph summarizing previous neuroscientific work. It would be more effective to end with a concise statement of the research gap and the paper’s explicit contributions, which are currently missing.
    - Structure of sections: The manuscript does not clearly distinguish between sections. For instance, the discussion still contains elements of methods and results. I recommend creating a dedicated subsection on statistical analyses or hypothesis testing within the Methods section, and expanding the Results section (currently only a short paragraph) to include descriptions of figures and main findings. The Discussion should then focus solely on interpreting these results and outlining their broader implications.
    - Discussion organization: The Discussion is organized according to the proposed hypotheses explaining why post-hoc selection is often neglected within systems neuroscience as according to the author. However, the connection between these hypotheses, the corresponding experiments, and the subsection titles is not always clear. For example, the title of subsection 4.2 does not reflect the context of model comparison mentioned in the hypotheses or shown in Figure 3. In contrast, subsection 4.3 presents a clearer alignment between subtitle and experimental results, but its link to the overarching hypothesis remains weak (and the hypothesis itself may not be directly falsifiable).

    Overall, improving the structural coherence and explicitly aligning hypotheses, experiments, and interpretations would greatly strengthen the manuscript’s clarity and scientific narrative.
4. The overall tone of the paper can be read as quite dismissive towards previous neuroscientific work. For example figure titles like ‘the illusion of progress’ (Fig. 3) or ‘why bother with veracity when volatility is king?’ (Fig. 4), as well as expressions in the ethics statement referring to “faulty methods,” “false results,” and “time wasted,” could be interpreted as overly critical. The paper would benefit from adopting a more neutral and descriptive tone throughout, focusing on constructive critique rather than evaluative language.

[1] Cawley, G. C., & Talbot, N. L. (2010). On over-fitting in model selection and subsequent selection bias in performance evaluation. The Journal of Machine Learning Research, 11, 2079-2107.

[2] Wainer, J., & Cawley, G. (2021). Nested cross-validation when selecting classifiers is overzealous for most practical applications. Expert Systems with Applications, 182, 115222.

[3] Guggenmos, M., Sterzer, P., & Cichy, R. M. (2018). Multivariate pattern analysis for MEG: A comparison of dissimilarity measures. Neuroimage, 173, 434-447.

**Questions:**

- The relationship between the repeated-stimulus confound and post-hoc selection bias remains unclear. How exactly is the repeated-stimulus confound defined in this context, and in what way does it interact with or contribute to post-hoc model selection bias?
- Could the authors provide more details about the dataset used, including the number of conditions, electrodes, and participants? Was this dataset selected because of its use in Kilgallen et al. (2025)? Furthermore, to what extent can the findings be generalized to other large-scale neural datasets, such as [4]?
- Could the authors clarify how EEG decoding was performed? Specifically, is the classifier trained separately for each time point, with performance averaged across time points and conditions, or is the neural data first averaged over time? Additionally, is a separate classifier trained for each participant, or is classification performed on data pooled across participants? The current methods section does not make these distinctions clear.
- Could the authors clarify how the hypothesis tests were conducted (in context of Table 3), given that the bias appears to be represented by a single value per model? A one-tailed t-test requires a distribution of values rather than a single estimate, so it is unclear what sample the test was performed on. Was the bias computed across cross-validation folds or participants?
- Could the authors elaborate on the hypothesis presented in Section 4.3 regarding the continued use of post-hoc model selection? It is unclear how this hypothesis could be falsified, and consequently, the connection between the hypothesis and the results is unclear.
- Could the authors clarify whether the range of overestimation in decoding accuracy (0.74–1.24%) mentioned in the abstract and Section 4.1 corresponds to the selection bias values reported in Table 3? In Table 3, the values appear to range from 0.14–1.44%; thus table and text seem to be inconsistent.

[4] Gifford, A. T., Dwivedi, K., Roig, G., & Cichy, R. M. (2022). A large and rich EEG dataset for modeling human visual object recognition. NeuroImage, 264, 119754.

---

> ### Author Response · Authors · 2025-11-21
> **Response to oCqR**
>
> Most of the main concerns and issues brought up have are addressed in the GENERAL COMMENTS above.
>
> Thank you for the references. Those are useful, and we'll incorporate them into the manuscript.
>
> ## W2
>
> The issue of overfitting to a dataset as a community has only one real solution: novel datasets. There is a rich literature on how to choose hyperparameters correctly, but we felt that to be beyond our scope here, which is to carefully identify and quantify and analyze the problem, rather than to propose a specific solution.
>
> ## W3
>
> Thanks for identifying these weaknesses in the presentation. We will edit the manuscript to address all of them, following your specific suggestions where possible.
>
> ## Q3
>
> Decoding accuracy ($Z_{i,j,k}$​) was tracked for each model ($i$), on the $k$-th fold, for the $j$-th subject.
>
> ## Q4
>
> The single selection bias value reported in Table 3 for each model and task represents the overall estimate, but the hypothesis test (t-test) was performed using the distribution of results across our experimental procedure.
>
> - Test Objective: We performed hypothesis tests to determine if the bias in decoding accuracy for each model was significantly greater than zero
>
> - Procedure: We applied one-tailed t-tests with a confidence level α=0.05.
>
> - To account for multiple tests, we applied the Holm-Bonferroni procedure at the task level
>
> - Sample Distribution: the accuracy measure was indeed derived from the 12-fold nested cross-validation procedure, and subsequent analysis utilized variance across subjects and folds. The sample distribution used for the t-tests reported in Table 3 was computed across the cross-validation folds.
>
> ## Q5
>
> The "hypothesis" discussed in §4.3 is the belief that post-hoc model selection is a relatively harmless practice in the broader context of the literature
>
> Connection to Results and Refutation:
>
> - We presented evidence that contradicts this belief
>
> - Our results showed that post-hoc model selection imparts a greater bias to unstable model architectures
>
> - Specifically, as instability (controlled by β) increases, the true accuracy of a model decreases, but the selection bias increases
>
> - We argue that the process is insidious because, since accuracy overestimation is proportional to instability, the publication process (which requires demonstrating improvement over prior solutions) implicitly encourages researchers to develop increasingly complex and unstable models that do not necessarily outperform simpler solutions.
>
> The empirical evidence that the bias is dependent on model instability, demonstrated using the Unstable-LR model (Figure 4), serves as the refutation of the notion that this practice is harmless
>
> ## Q6
>
> The manuscript is consistent: there is no contradiction between the text and Table 3.
>
> The range of overestimation reported in the Abstract and §4.1 (**0.74–1.24%**) refers specifically to the selection bias observed in the **RSC concept-decoding task**. We focused on this task because the affected publications were also susceptible to the repeated-stimulus confound (RSC), and the bias range (0.74–1.24%) is directly comparable to the historical margin of improvement in that literature (0.36–1.59%).
>
> The wider range you observed (**0.14–1.44%**) represents the minimum (0.14, LR model) and maximum (1.44, AW1DCNN model) selection bias values found across all three distinct tasks reported in Table 3 (Concept decoding, RSC concept decoding, and Stimulus decoding).

---

> > ### Comment · Reviewer_oCqR · 2025-11-27
> >
> > Thank you for your reply; this provides some clarification. However, I stand by my assessment that the novelty and overall contribution are too limited. In addition, to the best of my knowledge, the manuscript has not (yet) been updated to reflect these clarifications or to address the identified logical impasses. For these reasons, I will not raise my score.

---

### Author Response · Authors · 2025-11-21
**General Comments and Shared Concerns**

The novel contribution of this paper is *not* that hyperparameter search on the test set is problematic. That is well known. (We will incorporate the references helpfully suggested by oCqR.) The novel contributions are:

(a) That this confound is actually prevalent in a swath of recently published papers. (This is why the present manuscript is significant enough to merit publication: if a paper is invalid due to a confound, people must be able to discover that by searching the literature. The primary scientific literature should be self-correcting. This work is part of that self-correction process.)

(b) Careful measurement of the actual magnitude of the overestimate of performance due to this confound, across various models and datasets.

(c) Analysis of how the magnitude of the overestimate of performance is changing with time, leading to identification of a novel issue: that exploiting this confound is leading the community towards worse (less stable) models, and to an increase of the overestimate with time.

We will edit the manuscript to make this very explicit up front. We will also address other weaknesses, mainly in the presentation, pointed out in the reviews. And we will also add pointers to other works exploring occurrence of this confound, in other domains, to a "related work" section.

We are unsure about exactly how strong our tone regarding affected work should be. The confound is well known. As a community, we **should** be upset when work that is flawed in a well-understood way is published, particularly when it is a swath of work consuming enormous resources and misleading us about the true state of the art. Sometimes the pursuit of truth can be brutal, and we should not mince words when there is a problem. To make an analogy: if a swath of biochemistry results used a known-flawed assay, the follow-on critique exposing the issue might be quite strongly worded. (Or if sufficiently egregious and high-profile, even an indictment, trial, and prison.)

That said, we'll try to tone it down a bit.

To answer a few specific questions:

- The repeated-stimulus confound and the post-hoc model selection confound are unrelated: they are two different confounds. The only similarity is that they may each result in a community moving in the wrong direction: towards models that overfit to the specific image rather than the actual class for the first, and towards less stable models for the second.

- Details of the algorithms are in the supplementary material, and in the cited papers. But EEG is not averaged over time: the models take the signal as a function of time and location as input, perhaps temporally down-sampled and filtered.

- The EEG datasets in use are not unique to this study; details accompany the datasets, which can be freely downloaded.

- We chose this specific domain because of its scientific importance, and because hyperparameter optimization on the test set seems common there. A survey across many application domains checking for prevalence of this issue would be interesting, but is outside our scope, and would be a significant endeavour.

---

### Meta-Review · Area_Chair_a3uy · 2026-01-04

**Summary:**

The paper shows that post-hoc model selection (i.e. optimizing hyper-parameters on the test set) is problematic in the context of EEG decoding studies. The authors identify several existing works in this area that may have resorted to post-hoc model selection, and demonstrate that some of them show inflated performance. Then, the authors debunk some common misconceptions about post-hoc model selection, e.g. that the impact is not significant, that it doesn't affect the ranking of the methods. Then the authors show that post-hoc model selection encourages building unstable models, which can be detrimental long-term to the field.

All of the reviewers voted for rejecting the paper pre-rebuttal. The reviewers highlighted the following limitations:
- The paper presents a problem that is well-known and is commonly taught in machine learning classes: you cannot do model selection on the test set. The experiment in Figure 1 should be fairly clear to anyone who thought about model selection, for example.
- The paper doesn't propose any novel methods or solutions to the problem. Instead, it identifies and describes the issue.
- The paper is targeting a narrow subfield (EEG decoding)

One of the questions that was raised in the discussion is whether the paper is in-scope for ICLR. Generally, I believe that meta-studies and critiques of methodology in prior work may be in-scope. However, in its current form, the paper did not convince the reviewers that it is addressing an important issue, and that its results would be interesting to the ICLR community. From the machine learning perspective, the results are relatively clear and expected.

**Reviewer Concerns:**

- Unclear contribution, scope
  + discussed by the authors, partially addressed; the authors will make it clear that the paper is about a specific subfield (EEG decoding)
- No proposed solution to the issue
  + Not addressed, but in general a solution is not required for publication
- Limited novelty, presented results are clear from a machine learning perspective
  + Discussed, but generally not addressed

**Reviewer Scores:**

- Reviewer oCqR: 2 -> 2, major concern stands
- Reviewer LYdh: 2 -> 2, concerns discussed but not addressed
- Reviewer uDoR: 4 -> 4, concerns discussed but no clear case for increasing the score

---

### Decision · Program_Chairs · 2026-01-26

Reject